# Passive frequency comb generation at radiofrequency for ranging applications

Hussein M. E. Hussein [1,2,5], Seunghwi Kim [3,5], Matteo Rinaldi [1,2], Andrea Alù [3,4] ✉ & Cristian Cassella [1,2] ✉

Optical frequency combs, featuring evenly spaced spectral lines, have been extensively studied and applied to metrology, signal processing, and sensing. Recently, frequency comb generation has been also extended to MHz frequencies by harnessing nonlinearities in microelectromechanical membranes. However, the generation of frequency combs at radio frequencies (RF) has been less explored, together with their potential application in wireless technologies. In this work, we demonstrate an RF system able to wirelessly and passively generate frequency combs. This circuit, which we name quasi-harmonic tag (qHT), offers a battery-free solution for far-field ranging of unmanned vehicles (UVs) in GPS-denied settings, and it enables a strong immunity to multipath interference, providing better accuracy than other RF approaches to far-field ranging. Here, we discuss the principle of operation, design, implementation, and performance of qHTs used to remotely measure the azimuthal distance of a UV flying in an uncontrolled electromagnetic environment. We show that qHTs can wirelessly generate frequency combs with μWatt-levels of incident power by leveraging the nonlinear interaction between an RF parametric oscillator and a high quality factor piezoelectric microacoustic resonator. Our technique for frequency comb generation opens new avenues for a wide range of RF applications beyond ranging, including timing, computing and sensing.

Unmanned vehicles (UVs) have attracted growing attention in the past decade, generating new opportunities in a variety of emerging Internet of Things (IoT) applications, including precision farming[1], aerial imaging[2], smart manufacturing[3], maintenance in harsh locations[4], first response, and assisted living[5,6]. However, for UVs to play a key role in all these applications, it is necessary to precisely localize their positions in a variety of operational settings without compromising their battery life. In this regard, the global positioning system (GPS)[7] has been a key resource in the last decades for navigation and localization in outdoor settings. Nevertheless, GPS is often unavailable in indoor and underground environments, and its localization accuracy may exceed the size of today's UVs by orders of magnitude. Consequently, growing interest has been devoted to alternative methodologies for the localization of UVs in GPS-denied environments. In this regard, ranging techniques based on Light-Detection-and-Ranging (LIDAR)[8], ultrasound detection[9] and frequency-modulated-continuous-wave (FMCW) radars[10] have been extensively investigated, offering enhanced localization capabilities. Nonetheless, these techniques are power-hungry, making them unsuitable for low-power applications. Also, they require complex designs for the interrogation nodes, in addition to sophisticated pattern recognition algorithms[11].

[1]Department of Electrical and Computer Engineering, Northeastern University, Boston, MA 02115, USA. [2]Institute of NanoSystems Innovation, Boston, MA 02115, USA. [3]Photonics Initiative, Advanced Science Research Center, City University of New York, New York, NY 10031, USA. [4]Physics Program, Graduate Center, City University of New York, New York, NY 10016, USA. [5]These authors contributed equally: Hussein M. E. Hussein, Seunghwi Kim. ✉e-mail: aalu@gc.cuny.edu; c.cassella@northeastern.edu

Driven by the on-going Radio-Frequency-Identification (RFID)[12] revolution, increased attention has been also paid lately to ranging techniques based on radiofrequency (RF) passive tags. In particular, the adoption of RF passive tags has been recently proposed for localizing UVs in GPS-denied environments through a low-cost monitoring system not requiring any power from the targeted UVs[13–16]. A directional wireless transceiver can interrogate an RF passive tag onboard of a UV with a continuous-wave (CW) signal and leverage the received signal strength indicator (RSSI)[17] or the phase[18] of the backscattering signal to retrieve the distance between the tag and the transceiver. However, the RSSI of passive tags is inevitably distorted by multipath interference affecting the backscattered signal, which can be strong in indoor or underground settings as shown in Fig. 1a[19]. Such interference causes ranging errors that can become severe, especially when electrically small passive tags are used to fit into compact UVs. Relying on the phase difference between the received backscattered signal and the interrogation signal also leads to a low-ranging accuracy due to multipath, as well as to cycle-ambiguity[20]. Cycle-ambiguity arises because multiple distances between the tag and the transceiver can yield to the same phase difference between interrogation and backscattered signals[21]. Furthermore, since conventional passive tags operate in the linear regime, their backscattered signals have the same frequency as the interrogation signals. As a result, significant ranging inaccuracies can also be caused by electromagnetic clutter and by self-interference at the reader[22] as in Fig. 1b, c. In this context, a new category of tags operating in the nonlinear regime and known as harmonic-tags (HTs) has been recently proposed for ranging applications[13,21,23,24]. Unlike linear passive tags, HTs can employ the nonlinearities of varactors or diodes to generate backscattered signals at twice the frequency of the interrogation signal as in Fig. 1d, e. This feature provides HTs' readers with an immunity to both electromagnetic clutter and self-interference, beyond the limits of conventional linear passive tags[23]. Yet, the accuracy that a reader can ultimately achieve when remotely monitoring its distance from an HT in an indoor or underground setting remains inevitably limited by the multipath interference[24] affecting the HT's backscattered signal (see Fig. 1a)[19]. In fact, the only way for a reader to accurately assess its distance from an HT is to rely on power-hungry wideband transmitters and on intense signal processing[21]. Therefore, a new class of passive tags is needed to overcome the limits of existing counterparts used for ranging, enabling an accuracy insensitive to multipath interference affecting its backscattered signal, as well as to readers' self-interference and clutter.

Frequency combs have been extensively studied in the last 20 years as they provide robust and equally spaced spectral comb lines that can be used as frequency references. Frequency combs can be generated in nonlinear optical systems through mode-locking[25], Kerr nonlinearities[26,27] and electro-optic modulation[28,29]. In microresonator-based frequency combs, which are amenable for integration, the comb spacing is determined by the free spectral range of a certain resonator, and its width extends over GHz levels or even THz levels when operating at optical frequencies[30]. Such frequency synthesizers have opened several opportunities in the field of metrology, holding the promise for extremely precise timing and sensing applications, as well as for optical ranging[31,32]. Particularly, dual-comb systems can provide unprecedented ranging resolutions at fast rates. However, they typically rely on power-hungry free-space lasers incident on targets, and they are susceptible to large path losses, scattering losses and beam dispersion[33,34]. Frequency combs in the microwave range have also been recently demonstrated through strong light-matter interactions[35] in optical microresonators, as well as in microelectromechanical systems (MEMS) utilizing nonlinear three-wave mixing[36–39], nonlinear friction forces[40,41] and strong nonlinear electromechanical couplings[42]. References [43,44] recently demonstrated comb generation in Josephson-Junction circuits through three-wave mixing for quantum engineering

applications. However, to our knowledge, there have been no attempts to generate and employ frequency combs in passive RF systems for ranging applications. To this end, simultaneously leveraging the strong nonlinearities of solid-state components and the large quality factor ($Q>1,000$) of microacoustic piezoelectric resonators[45–48] offers an ideal platform for low-power comb generation in the RF range.

In this Article we demonstrate a nonlinear RF passive tag engineered to realize comb-based ranging with accuracy intrinsically immune to both self and multipath interference affecting its backscattered signal. Such a passive tag, referred to as a quasi-Harmonic Tag (qHT), receives an interrogation signal with frequency $\omega_p$ and power $P_r$, and it responds by passively generating a frequency comb through its nonlinear behavior, as shown in Fig. 1f. The comb is symmetrically distributed around half the frequency of the interrogation signal (i.e. $\omega_p/2$), with a comb line spacing ($\Delta f$) that is a function of $P_r$ as in Fig. 1g. This differs from frequency comb generation in optical systems, where the comb-line spacing remains independent of the input power level. The inverse proportionality between received power ($P_r$) and its distance from the interrogating node ($d$), described by the Friis transmission relation plotted in Fig. 1h[49], can be exploited by qHTs to retrieve $d$. This can be done by simply measuring the comb line spacing, as sketched in Fig. 1i. In this regard, any measured $\Delta f$ value univocally maps onto one specific $P_r$ level, and consequently to a specific value of $d$. Therefore, readers with a directional transmitter can remotely assess their distance from a qHT by simply extracting $\Delta f$, and the accuracy of such extraction is not degraded by multipath. In fact, undesired scattering caused by multipath can distort both the RSSI and the phase of the current RF passive tags' backscattered signal, but they have no effect on $\Delta f$ in qHTs. Moreover, qHTs' ability to utilize separate bands for receiving the interrogation signal and transmitting their backscattering signal confers immunity to electromagnetic clutter. Also, it enhances the resilience of qHTs' readers to their own self-interference by allowing them to filter all the harmonics of the transmitted interrogation signals that are generated due to nonlinearities in the power amplifier of their transmitter. In turn, a minimum level of received power, designated as $P_{th}$, is required to activate the generation of a frequency comb in qHTs. In this regard, the value of $P_{th}$ ultimately sets the maximum distance from which qHTs can be remotely interrogated.

A typical application scenario leveraging the $\Delta f - P_r$ dependence characterizing the operation of qHTs is sketched in Fig. 1j, where we envision a qHT mounted onboard of a drone that is sequentially interrogated by different *beacons* located at predetermined positions. The qHT generates unique frequency comb patterns corresponding to its distance $d$ from each interrogating beacon. Hence, by extracting $\Delta f$ from each one of the beacon's received signals, and by using trilateration, we can identify the azimuthal position of the qHT within the common area covered by the beacons.

## Results

### Model: frequency comb generation in qHTs

As mentioned previously, a fundamental characteristic of qHTs is the generation of frequency combs with line spacing that varies according to the received power level. This feature allows readers to accurately determine their distance from a designated qHT. Consequently, it is important to ascertain the key parameters that control the comb generation in a qHT, as these parameters play a pivotal role in the design of these tags. With no loss of generality, a qHT can be described as a parametric oscillator (PO)[50], coupled to a high quality factor ($Q$) piezoelectric microacoustic resonator with resonance frequency matching or closely matching the PO's output frequency. We can describe this system with a toy model formed by two electrical modes and a mechanical mode, as shown in Fig. 2a. The two electrical modes, namely the signal mode $a_1$ and the pump mode $a_2$, are parametrically coupled via a nonlinearity, while the mechanical mode $b$ is linearly

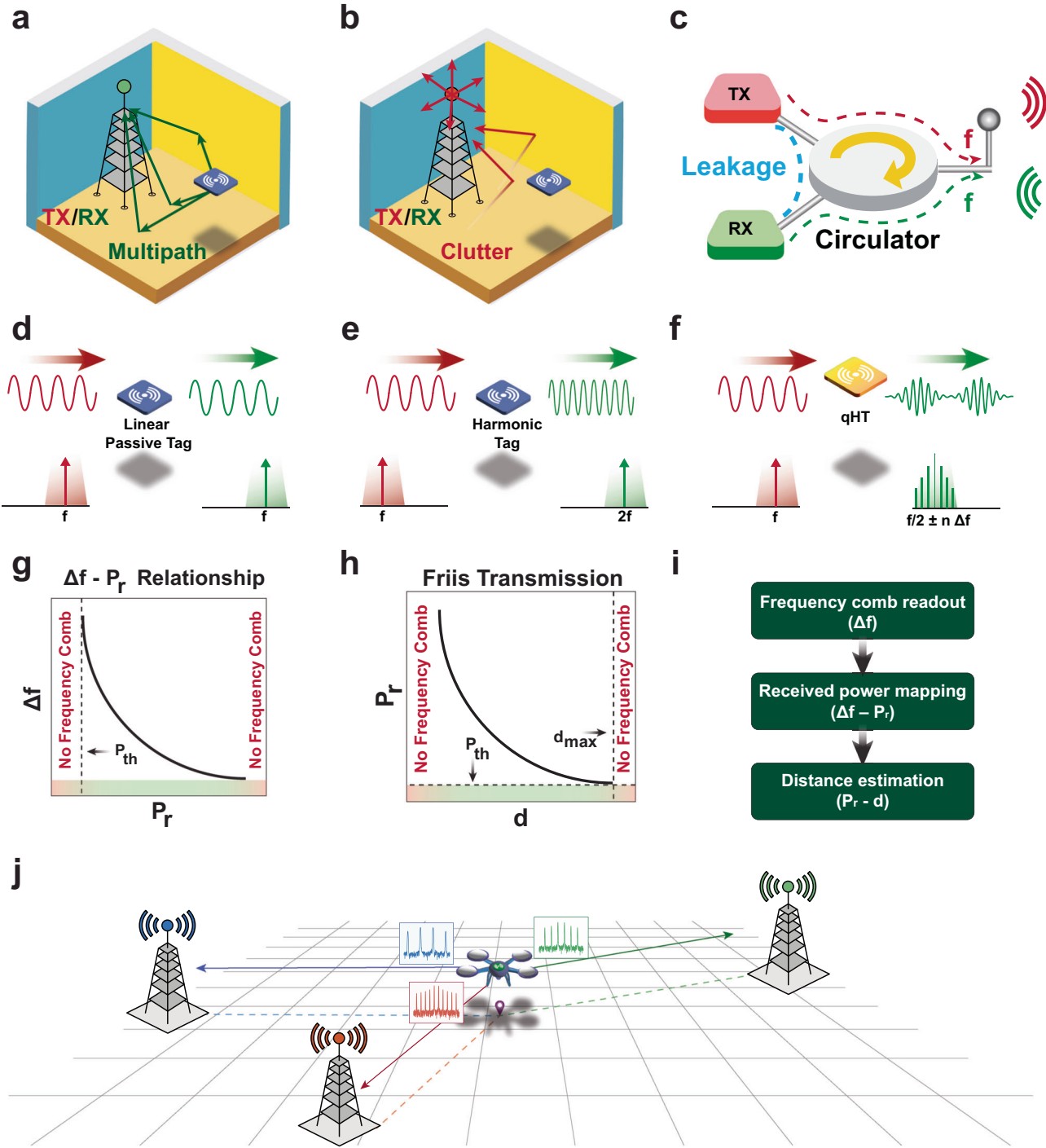

**Fig. 1 | Overview of challenges and proposed solution. a** Miniaturized RF passive tags radiate their backscattered signals omnidirectionally. Consequently, their readers are prone to loss of accuracy due to multipath interference, especially when operating in rich-multipath settings. **b** The receiver of RF transceivers may experience loss of accuracy due to clutter when remotely sensing a parameter of interest through a linear RF passive tag. TX: transmit, and RX: receive. **c** RF transceivers using the same frequency channel for both transmission and reception may suffer from loss of accuracy due to self-interference (SI) when sensing a parameter of interest through a linear passive tag; SI may be caused by limited isolation in the circulator that such transceivers generally leverage to be able to transmit and receive information with the same antenna. **d**–**f** Schematic representation

describing the spectral characteristics of interrogation and backscattered signals for linear RF passive tags, harmonic tags and quasi-harmonic tags (qHTs). **g** Dependance of qHTs' comb line spacing $\Delta f$ on their received power level $P_r$. **h** Dependence of $P_r$ on the distance $d$ of a qHT. **i** Flowchart illustrating the required steps to read-out $d$ from $\Delta f$. **j** A typical application scenario for qHTs, wherein the position of a qHT onboard of a drone is resolved by extracting the $\Delta f$ values produced by the qHT after interrogating it sequentially with three separate beacons located at three known positions. The extraction of the drone distance from each beacon allows the localization of the drone with high accuracy, even in GPS-denied settings.

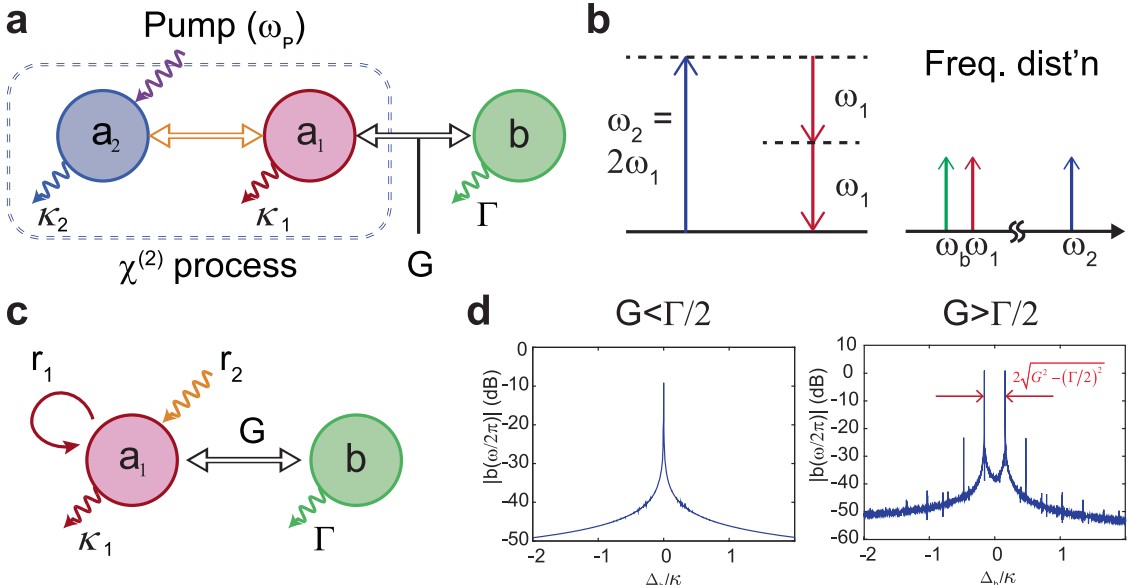

**Fig. 2 | Analytical toy model of the qHT. a** Schematic of the analytical model we developed to identify the origin of the frequency comb generation in qHTs. The toy model consists of triply resonant modes, with two electrical modes ($a_1$ and $a_2$) coupled by nonlinearity via the chi-2 process, and the signal mode $a_1$ coupled to a mechanical mode $b$. The impinging pump is resonant at the pump mode $a_2$. **b** Energy levels for the chi-2 process, where the signal mode $a_1$ is indirectly excited by the pump via the parametric down conversion $\omega_2 = 2\omega_1$, and the frequency of the $b$ mode is close to the frequency of the $a_1$ mode ($\omega_b \approx \omega_1$). These relationships are visually illustrated on the right in the frequency distribution. **c** Schematic of the 'simplified model' used in our analytical treatment, where the $a_1$ and $b$ modes are

'linearly' coupled via the electromechanical coupling rate G. The $a_1$ mode undergoes additional nonlinearities, including the gain saturation ($r_1$) and small-signal gain ($r_2$) terms due to the parametric process. **d** Frequency spectrum for the weak (left) and strong coupling (right) regimes at Re[$s$] = 0 obtained from our numerical analysis. Frequency-combs are only observed when the coupling is larger than the mechanical loss rate $\Gamma$, and the comb spacing right after the comb emerges is given by $\Delta f_{max} = 2\sqrt{(G^2 - (\Gamma/2)^2)}$, which matches well with the maximum comb spacing from our numerical analysis.

coupled to the signal mode with electromechanical coupling rate $G$[51,52]. The pump mode $a_2$, is driven by a microwave pump signal at $\omega_p$, which indirectly excites the signal mode $a_1$ through parametric down-conversion (Fig. 2b-left). This interaction is made possible by a second-order nonlinearity ($\chi^{(2)}$), leading to $\omega_2 = 2\omega_1$, where $\omega_1$ and $\omega_2$ denote the natural resonant angular frequencies of $a_1$ and $a_2$, respectively[53]. Since the loss rate of $a_2$ is larger than the one of $a_1$, we adiabatically eliminate the pump mode $a_2$ from the equations of motion, reducing them to a single equation for the nonlinear interaction (see detailed analysis in Supplementary Section I). The mechanical mode, with resonant frequency close to the frequency of $a_1$ ($\omega_b \approx \omega_1$), is linearly coupled to $a_1$ as shown in (Fig. 2b-right). Hence, we can rewrite the equations of motion as shown in Fig. 2c:

$$\dot{a}_1 = -\left(\frac{\kappa_1}{2} + i\Delta_a\right)a_1 - iGb - r_1|a_1|^2 a_1 - ir_2 a_1^*,$$
$$\dot{b} = -\left(\frac{\Gamma}{2} + i\Delta_b\right)b - iGa_1, \quad (1)$$

where $r_1$ is the gain saturation coefficient and $r_2$ is the small-signal gain due to parametric amplification. Additionally, the detuning coefficients are given by $\Delta_a = \omega_1 - \omega_p/2$ and $\Delta_b = \omega_b - \omega_p/2$, while $\kappa_1$ and $\Gamma$ are the loss rates of $a_1$ and $b$; hence, the mechanical quality factor is $Q \equiv \omega_b/\Gamma$. We can simplify Eq. (1), combining them into a single equation by integrating the coupling term:

$$\dot{a}_1 = -\left(\frac{\kappa_1}{2} + i\Delta_a\right)a_1 - r_1|a_1|^2 a_1 - ir_2 a_1^* + F(t), \quad (2)$$

where $F(t) \equiv -G^2 \int_0^t d\tau f(\tau) a_1(t-\tau)$ represents the external stimuli applied to $a_1$, and $f(\tau) \equiv e^{-\left(\frac{\Gamma}{2} + i\Delta_b\right)\tau}$ is the linear response function of the mechanical mode[43]. The stimuli $F(t)$ plays a crucial role in the

generation of frequency combs, with comb generation contingent upon the strength of the coupling rate $G$ as discussed below.

In the weak electromechanical coupling rate regime ($G \ll \Gamma/2$), the last term in Eq. (2) simplifies to $F(t) \approx -\frac{G^2}{\Gamma/2 + i\Delta_b} a_1(t)$. Thus, Eq. (2) is characterized by an effective loss $\bar{\kappa}_1 = \kappa_1 + \frac{G^2}{(\Gamma/2)^2 + \Delta_b^2}\Gamma$ and an effective detuning $\bar{\Delta}_a = \Delta_a - \frac{G^2}{(\Gamma/2)^2 + \Delta_b^2}\Delta_b$. Linear stability theory can be utilized to further analyze Eq. (2) by using the perturbative series $a_1(t) \rightarrow \alpha + \delta a_1(t)$ to derive the pole of the system (see Supplementary Section II). For large pump power levels, i.e., $r_1^2|\alpha|^4 + r_2 > \bar{\Delta}_a^2$, the real part of the pole is Re[$s$] $= -\frac{\bar{\kappa}_1}{2} - 2r_1|\alpha|^2 \pm \sqrt{\left(r_1^2|\alpha|^4 + r_2\right)^2 - \bar{\Delta}_a^2}$ while the imaginary part is zero. Consequently, a single-frequency oscillation is observed at $\omega_p/2$, as shown in Fig. 2(d-left). The synchronization of the signal mode $a_1$ at half frequency of the pump results from the typical response of degenerate parametric systems, which do not exhibit multiple oscillations at frequencies uncorrelated from their pump frequency[50]. Thus, frequency combs cannot be generated by our system for $G \ll \Gamma/2$.

On the other hand, when the system is operated in the strong coupling regime ($G > \Gamma/2$), the investigation of its stability requires applying the Laplace transform to Eq. (2) after its linearization. This allows to verify that frequency combs are generated at Re[$s$] = 0, with a maximum comb spacing $\Delta f_{max} = 2\sqrt{G^2 - (\Gamma/2)^2}$ obtained when the pump reaches the threshold for comb generation (i.e., when $P_r = P_{th}$). Such combs cannot be observed in the weak coupling regime. Figure 2d illustrates the numerically evaluated spectrum of $a_1$ at Re[$s$] = 0 where the comb starts emerging, and the analytically calculated maximum spacing is remarkably close to the corresponding

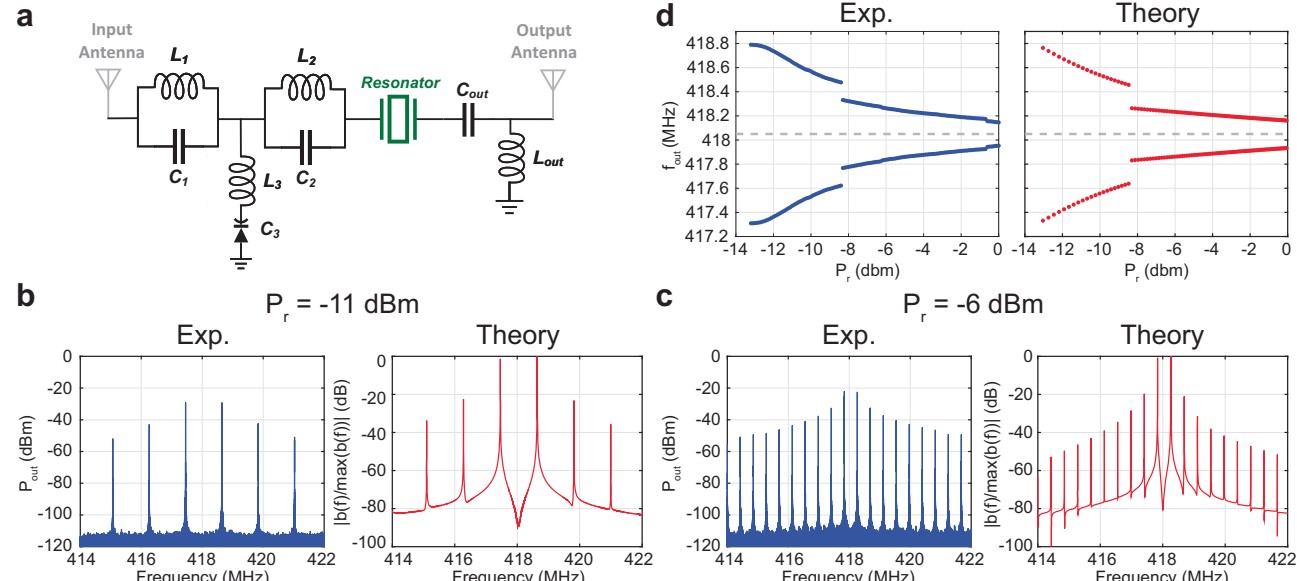

**Fig. 3 | Measured and analytical results of the qHT. a** Circuit schematic of the qHT built in this work. **b, c** Measured (in blue) and numerically predicted (in red) output spectrum of the qHT built in this work for two different input power levels. **d** Measured (in blue) and numerically predicted (in red) frequency $f_{out}$ for the first two tones around half of the adopted interrogation frequency (836 MHz)

generated by the built qHT for different input power levels. The inductors $L_1, L_2, L_3$ and $L_{out}$ were selected to have inductances equal to 61 nH, 63 nH, 36 nH and 3.6 nH, respectively. The capacitors $C_1, C_2, C_3$ and $C_{out}$ were selected to have capacitances equal to 2.3 pF, 0.6 pF, 2.3 pF and 103 pF, respectively.

numerically extracted value. By increasing the small signal gain $r_2$, the pole can be further pushed above the real axis, causing the comb spacing to decrease and progressively driving the system into a synchronized period-doubling regime. In other words, the synchronization is frustrated for a wide range of $r_2$ under the strong coupling regime[54,55] (See detailed discussion in Supplementary Material Section II). In this range, frequency combs are observed and the comb-line spacing is a monotonic function of $r_2$, which is the key feature to produce ranging measurements without ambiguities. More details on our analytical model are provided in Supplementary Section II.

## Experimental results

To showcase the functionalities of qHTs, we have developed a prototype based on off-the-shelf components assembled onto a printed circuit board (see Fig. 3a). This qHT consists of an input mesh and an output mesh, each containing a set of lumped electronic components and one antenna. The two meshes share a solid-state varactor and an inductor ($L_3$), forming a two-port degenerate parametric circuit for subharmonic generation. The input mesh has an LC-notch filter with a resonant frequency $\omega_1/2\pi = 1/\sqrt{L_1 C_1} \approx 418$ MHz, which is approximately half of the resonant frequency of the LC notch filter placed in the output mesh ($\omega_2/2\pi = 1/\sqrt{L_1 C_1} \approx 836$ MHz). The presence of these two LC-filters ensures that the interrogation signal (i.e., the "pump"), with angular frequency equal to $\omega_2$, is fully confined in the input mesh, while its parametrically generated output signal, at or around $\omega_1$, is constrained in the output mesh. In addition, a lumped impedance transformation stage, consisting of an inductor ($L_{out}$) and a capacitor ($C_{out}$), is included in the qHT's output mesh to reduce the impact of the antenna loading on the lowest achievable $P_{th}$ value.

As described in the previous section, the generation of frequency combs in qHTs is enabled by the introduction of a high-$Q$ piezoelectric microacoustic resonator, with a natural resonant frequency $\omega_b$, coupled to the $a_1$-mode through the electromechanical coupling rate ($G = \sqrt{2}k_t^2\omega_b$, where $k_t^2$ is the electromechanical coupling coefficient[56,57], see Supplementary Material Section III). In this regard, the microacoustic resonator must be coupled to the $a_1$ mode strongly, which requires that its $k_t^2$ is larger than $\frac{1}{\sqrt{2}Q}$. In order to address this

requirement and enable frequency comb generation in our qHT, we employ a Surface Acoustic Wave (SAW) resonator in the qHT's output mesh. Such a SAW device has its resonant frequency at $\omega_b/2\pi \approx 418$ MHz and a $k_t^2 (\sim 10^{-3})$ larger than the inverse of its quality factor ($10^4$). Such $Q$ and $k_t^2$ values permit to enter the strong coupling regime required for comb generation, as further discussed in Supplementary Section II. It is worth mentioning that using a high-$Q$ resonator is essential for the generation of frequency combs in our system. In fact, as verified through simulations (see Supplementary Material Section III.E), it is impossible to generate a frequency comb in our system if the quality factor is lower than ~60 or the resonator's equivalent inductance is in the μH-range or lower (which is the case for any available inductors with self-resonance higher than 900 MHz). More details on the design procedure, on the component selection criteria and on the operation of the qHT are provided in Supplementary Sections II and III.

To characterize the electrical response of the qHT, we first performed a wired experiment where we directly connected the input and output ports of the device to a signal generator and a spectrum analyzer respectively (see Supplementary Section III). We set the operating interrogation frequency for frequency comb generation to 836 MHz, which is approximately twice the resonant frequency of the resonator. Afterwards, we identified $P_{th}$ (-13 dBm) by gradually increasing $P_r$. Extracting this value is crucial as $P_{th}$ directly relates to the read-range of the qHT, i.e., the maximum allowed distance of the qHT from its wireless transmitter. Then, we studied the dependance of the measured $\Delta f$ values on $P_r$, which is the key feature allowing to wirelessly and reliably extrapolate the distance of the qHT from a wireless interrogating node. To this end, we analyzed the measured output spectrum for different $P_r$ values (Fig. 3b, c), followed by the extraction of $\Delta f$ for the same range of $P_r$ values (Fig. 3d). Consistently with our theoretical analysis, we found $\Delta f$ to vary monotonically with $P_r$, which is a necessary condition to avoid ambiguities when correlating an extracted $\Delta f$ value to only one specific $P_r$ value and, consequently, to only one $d$ value. We also detected the presence of a subcritical bifurcation[58] for $P_r \approx -8.5$ dBm. This bifurcation is attributed to higher order nonlinearities in the varactor's capacitance vs. voltage characteristic, which are significant when

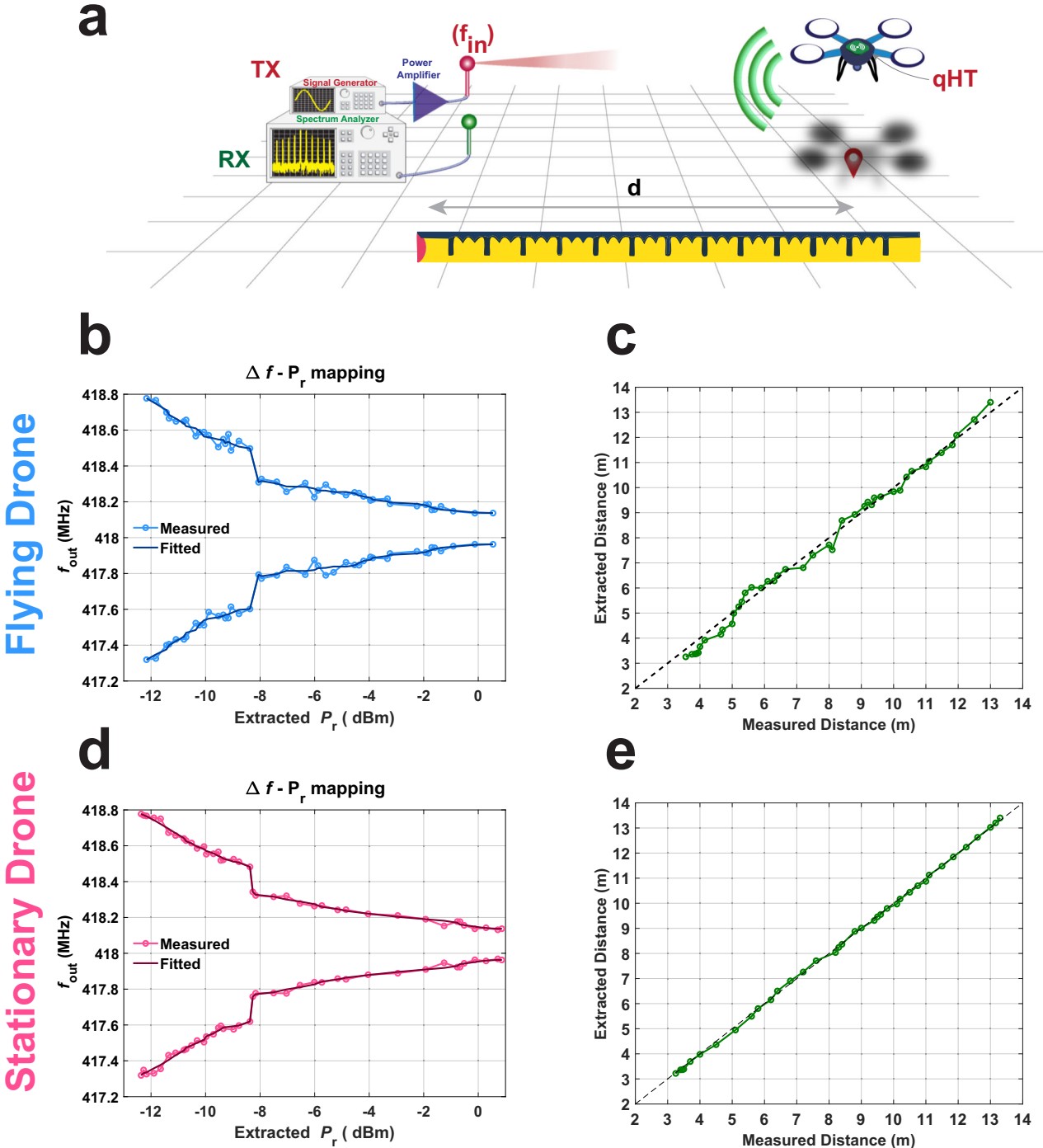

**Fig. 4 | Measured results of the qHT when localizing a drone. a** Schematic view of the experimental setup we used to assess the azimuthal distance of a drone at different distances from our interrogation node. The qHT is mounted on top of the drone and a signal generator connected to a power amplifier is used to remotely interrogate the qHT. The qHT responds with a frequency comb that is received by a spectrum analyzer. Measured frequencies for the strongest tones of the generated frequency comb *vs.* the corresponding extracted $P_r$ values when the distance is spanned by the drone while flying (**b**) or when the drone is manually moved (**d**). **c**, **e** The extracted drone's distance from the interrogating node vs. the distance measured with the ruler. The extracted distance is found by: (i) extracting $\Delta f$; (ii) using the extracted $\Delta f$ value to find $P_r$; (iii) using the found $P_r$ value and the Friis space propagation model to determine $d$. The reported values of $d$ have been extracted from $\Delta f$ for both the case in which the drone is flying (**c**) and the case in which it is manually moved (**e**).

the varactor exhibits a voltage across its terminal approaching its built-in potential ($V_{bj}$), as it occurs in this case. Further discussion regarding this subcritical bifurcation is provided in Supplementary Section III. It is worth emphasizing that the measured output spectrum of the built qHT closely matches what expected based on our analytical model, as illustrated in Fig. 3b–d.

After building the $\Delta f - P_r$ mapping plot (see Fig. 3d), we validated our proposed readout technique as sketched in Fig. 1f. We performed an experiment with the built qHT placed onboard of a drone flying in a hall of the Interdisciplinary Science and Engineering Complex (ISEC) at Northeastern University (see Fig. 4a). A signal generator was connected to a directional Yagi antenna (A1) to generate a CW

interrogation signal at $\omega_p$. Also, we relied on a spectrum analyzer connected to an off-the-shelf dipole antenna to wirelessly receive a portion of the backscattered signal generated by the qHT while remotely moving the drone along the direction of maximum gain of A1. A detailed description of the measurement setup is given in Supplementary Section III, together with a video of the performed experiment in Supplementary Video 1. During this experiment, we collected the received $\Delta f$ value for multiple points within the qHT's read-range while simultaneously measuring the actual distance by using a ruler. From these $\Delta f$ values, we extracted the corresponding $P_r$ values based on the $\Delta f - P_r$ mapping plot discussed in Fig. 3d, as shown in Fig. 4b. Finally, we evaluated the distance of the drone for each investigated position directly from the corresponding extracted $P_r$ value by leveraging the Friis transmission equation. All the extracted $d$ values are compared with those measured with the ruler in Fig. 4c. We found that the error between the actual measured distances and the distances extracted through the collected $\Delta f$ values does not exceed $\pm 57$ cm across a broad range of distances up to 13 meters. Interestingly, the maximum error measured at the farthest distance was found to be $\pm$ cm, which is less than the maximum error found across the entire measurement range. It is worth noting that the accuracy of our measurements has been degraded by the inherently unstable nature of the drone while hovering. To better assess the intrinsic accuracy of the qHT, we performed an additional ranging experiment wherein the drone was manually moved. In particular, we utilized the same setup used in the previous experiment, and we measured the distance of the drone after moving it manually to several stationary positions away from A1. We rebuilt the $\Delta f - P_r$ mapping plot extracted from the previous experiment in Fig. 4b, as shown in Fig. 4d. Because of the absence of the drone's hovering fluctuations, we were able to obtain a better accuracy, with a ranging error lower than $\pm 16$ cm across the same 13 meters operational range [see Fig. 4d, e]. Such measured accuracy is better than what achievable when relying on state-of-the-art passive tags[21,59–61] for measuring the distance of assets deployed >4 meters away from an interrogating node in an uncontrolled electromagnetic setting. Moreover, the qHT introduced in this work shows the lowest relative error among the passive tags demonstrated for far-field ranging (see Supplementary Table 3), equal to the ratio of the maximum error value to the maximum range value. The relative error demonstrated in this work is even lower than what previously achieved using state-of-the-art passive tags in an anechoic chamber. Also, our qHT just requires a single-tone interrogation signal. As a result, the demonstrated qHT allows employing readers equipped with narrow-band low-power transceivers for the interrogation, and enables an accurate extraction of ranging information without running intense signal processing operations. Furthermore, being able to extract ranging information by only using a single-tone interrogation signal, makes it possible to envision the interrogation of future qHTs through existing communication signals, a technique often referred to as "integrated communication and sensing"[62] that has becoming more and more popular in the last few years.

## Discussion

In this article, we have introduced quasi-Harmonic tags (qHTs), and we demonstrated their application for ranging measurements of a UV operating in a rich-multipath environment. Our demonstration encompasses both theoretical analysis and experimental validation, revealing a distinctive property: the comb-line spacing of qHTs exhibits an inverse relationship with the received input power level. This feature offers a valuable opportunity to achieve ranging capabilities with accuracy surpassing existing RF passive tags. In fact, by encoding the ranging information into the comb line spacing, and not into the amplitude or phase of the backscattered signals like in conventional RF passive tags, qHTs avoid ranging inaccuracies caused by multipath affecting their backscattered signal. Our experiments illustrate that it

is possible to measure the distance of a qHT located up to 13 meters away from a CW transmitter, in an uncontrolled electromagnetic environment, with an accuracy not exceeding $\pm 16$ cm. In the future, by recurring to a single-antenna circuit architecture[63] and to a micro-acoustic resonator exhibiting lower losses than the SAW device used in this work, we expect to be able to significantly extend the maximum achievable operational range beyond 13 meters. The ability to generate frequency combs in RF systems not only provides new ways to mitigate multipath distortion in wireless sensing systems relying on passive tags, but also opens exciting opportunities for leveraging RF frequency combs in next-generation timing, computing, and sensing systems.

## Methods

### Design and fabrication

We followed the design procedure we reported in[50] for designing parametric frequency divider circuits with the lowest possible power threshold $P_{th}$. With the proper optimization of the circuit around the frequency of operation of the pump signal at 836 MHz, the optimal values of inductors $L_1, L_2, L_3$ and $L_{out}$ are found to be equal to 61 nH, 63 nH, 36 nH and 3.6 nH, respectively. Meanwhile, the optimal values for the capacitance of the capacitors $C_1, C_2, C_3$ and $C_{out}$ are found to be equal to 2.3 pF, 0.6 pF, 2.3 pF and 103 pF, respectively.

We built the circuit on a printed circuit board (PCB) with a standard FR-4 substrate relying on off-the-shelf lumped components. The employed varactor diode used in our circuit is a skyworks diode with model number SMV1405, while the SAW resonator is a commercial Abracon resonator with resonance frequency of 418 MHz and model number ASR418S2. We connected commercial off-the-shelf antennas at the input and output of the circuit with model numbers AEACAC054010-S915 and 712-ANT-433-CW-QW, respectively.

### Comb measurements

In order to characterize the change of the generated frequency comb with respect to the change in the input power level reported in Fig. 3d, we connected a signal generator Tektronix TSG 4104 A to the input of our qHT to feed it with a CW signal at frequency 836.1 MHz while we monitored the output spectrum through connecting the output of the qHT to a signal analyzer Keysight N9010A EXA. We programmatically swept the signal generator power level and recorded the corresponding generated frequency comb at every single power level to generate the plot reported in Fig. 3d. For the drone experimental setup, we replaced all the cables with the antennas, and we connected a Yagi-Uda antenna to the signal generator and power amplifier ZHL-1000-3W + . The power level of the signal generator was fixed and mounted the qHT on the top of a drone and we manually swept the flying drone position away from the interrogating antenna while simultaneously recording the backscattered frequency comb response at every single position point.

## Data availability

All the data supporting the findings of this study are available within the main text and the Supplementary Information. The source data used in generating the measured results in Fig. 3 is provided in the Source Data file. All other data that support the findings of this study are available from the corresponding author upon request. Source data are provided with this paper.

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

## Acknowledgements

H.H., C.C. were funded by the National Science Foundation (NSF) under grant No. 1854573 and by the DARPA Optomechanical Thermal Imaging (OpTIm) program. S.K., A.A. were supported by the Air Force Office of Scientific Research and the Simons Foundation. The authors thank Sherif Badran, Nicolas Casilli and Davide Villa for their help with the experimental setup.

## Author contributions

C.C. conceived the idea and developed the research plan; H. H. designed and fabricated the device; S.K. and A.A. conducted the analytical calculations and theoretical modeling. C.C. and H.H. contributed to the design of the experiments; H.H. performed the experiments; H.H. and C.C. analyzed the data; C.C., A.A. and M.R. supervised the research; All authors contributed to writing the paper.

## Competing interests

The authors declare no competing interests.
