## [Peer Review File · Nature Communications]

Passive frequency comb generation at radiofrequency for ranging applicationsReviewer #1 (Remarks to the Author):

The manuscript titled "Passive frequency comb generation at radiofrequency for ranging applications" by Hussein M. E. Hussein, Seunghwi Kim, Matteo Rinaldi, Andrea Alù, and Cristian Cassella introduces a new type of RF passive tag called the quasi-Harmonic tag (qHT). This tag generates a frequency comb symmetrically distributed around half the interrogation signal frequency, and the comb's spacing is a function of the interrogation power. The results are interesting and would be beneficial for engineers designing devices for ranging measurements of UV in adversarial environments.

However, while the theoretical analysis is sound, and the experimental validation is appropriate, I am not convinced that this manuscript provides a clear advance in scientific understanding that would excite the interest of Nature Communications' readership. Therefore, I suggest that the authors target a specialized research journal, such as Nature Microsystems and Nanoengineering or Nature Communications Engineering, to find a more appropriate audience.

To enhance the manuscript's impact, I recommend that the authors answer the following questions:

1. What is new in the presented theoretical analysis that distinguishes it from countless publications studying the nonlinear coupling between a parametric oscillator and high-Q mechanical resonators?
2. How can the equations of motion shown in (1) be rewritten as equation (2), where the mechanical mode b is no longer present? Is the mechanical mode linearly or nonlinearly coupled to the signal mode a_1 ? There seem to be conflicting statements in line 147 and line 203 that require clarification.

Reviewer #2 (Remarks to the Author):

The authors describe a circuit that exploits frequency comb generation for ranging applications. Frequency combs have been extensively studied in optics and the theory and experimental demonstration of the technique described here was performed in references [43,44] using a Josephson junction circuit. The main innovation of the present work consists therefore in demonstrating an application to ranging by exploiting the relationship between the pump power in the circuit and the frequency spacing of the generated harmonics. The technique in particular uses a system of two coupled resonances, one nonlinear and one linear, which is a minimal system to generate frequency combs as detailed in ref [43]. The proposed solution is advantageous compared to linear antennas because it maps the target distance to a frequency difference between two tones, rather than a time delay. This is therefore potentially less susceptible to multipath interference and jamming. I have a few comments:

- 1) The weak coupling regime analysis assumes the condition $G < \Gamma/2$. However the Markovian approximation is technically for $G \ll \Gamma/2$ (much less, not simply less), i.e. for G close to $\Gamma/2$ I would not expect the Markovian approximation to still hold, even if this is still in the weak coupling regime. I think this should be clarified.
- 2) From the description on page 10 of the circuit in Figure 3a the authors say that L_1, C_1 resonates around $\omega_p/2$, but in the supplementary material, section III.a, they say that it resonates at ω_p . The latter makes more sense to me, since otherwise the pump would not be able to drive the circuit resonantly and the SAW resonator would not be strongly coupled to the electrical resonator at $\omega_p/2$.
- 3) What is the purpose of the third resonator L_3, C_3 ?
- 4) What is the intuitive reason why I should expect the distance between the comb frequencies to

decrease at higher pump power?

5) The condition for stability limits the maximum value for r_2 . How does this translate in the maximum allowed power? In principle a jammer could be designed that potentially disables the device by sending more power than the stability threshold, thereby reducing the effectiveness of the proposed technique.

6) In deriving equation S.2 in the supplementary from equation S.1 the authors perform a change of variables to move from the lab frame (where a_1 and a_2 oscillate at frequency ω_1 and ω_2) to a rotating frame (where the the variable a oscillates around Δ_a). This point should be clarified in the text in order to avoid confusion.

7) The authors state after equation S.4 that Δ_b is now assumed to be zero. However the formulas after equation S.5 assume $\Delta_b \neq 0$. I would suggest to clarify at which point in the derivation we should assume $\Delta_b=0$, maybe by moving this sentence to later in the text.

8) Did the authors perform EM liner or nonlinear simulations in the circuit design?

9) What is the measured Q factor of the SAW resonator?

10) The authors indicate that the ranging distance could be extended beyond the 13 meters demonstrated here, which are limited by the threshold power for the harmonic generation to start. In fact this is potentially a limitation compared to passive tags, which do not have a minimum threshold of operation. To understand the applicability of the proposed technique, can the authors comment on how can the distance be increased (and therefore the threshold power decreased)?

Reviewer #3 (Remarks to the Author):

The paper presents a nonlinear circuit based on lumped elements and MEMS resonators, which is capable of frequency comb generation at microwave frequencies. An important aspect of this circuit is the dependence of the frequency comb spacing on the driving pump power. The authors show an application of the circuit is RFIDs, specifically the determination of the location of a tag from the frequency spacing of the emitted comb. The analysis includes theoretical and experimental results in good agreement with each other. The work is generally interesting and innovative.

A major issue with the paper is somehow a lack of a cohesive message about its most important and novel result. The paper essentially deals with two topics, the design of circuits for microwave frequency comb generation and the use of such circuits for RFID applications. However, it is not immediately clear to what degree each of these topics advances their respective areas. For example, one of the most exciting aspects of the proposed circuit is the dependence of the comb frequency separation on the pump power. However, it seems that this effect has been theoretically predicted in Ref. 43. An experimental demonstration of this effect is very interesting, but it is not clear whether this is indeed the main contribution of the paper. Furthermore, it is unclear whether this is the first experimental demonstration of this effect or not. The use of MEMS resonators is another interesting aspect of the circuit, but it turns out that their role is auxiliary, since the nonlinearity that is responsible for comb generation is obtained through varactors. In principle, the circuit could be designed with conventional components only by following the architecture in Ref. 43, and the benefit of using MEMS resonators seems to be in controlling the decay rate of one of the cavities. The advantages of the proposed RFIDs from the state of the art are clearer, especially through Table S3 of the SI. However, it is unclear whether they present such a significant advancement to warrant publication in a broad-interest journal instead of a more specialized one. The authors need to spell out better the major contributions of their work and build a case for them.

Other than this general issue, the paper is written very well. The theoretical analysis is extensive and explains well multiple aspects of the proposed circuit. The experimental results are also very interesting and provide solid proof of the concept.

In summary, the paper is interesting, but the authors need to explain better its contribution to determine the suitability of their work for publication in Nature Communications.

Response to the Comments from the Referees on “Passive frequency comb generation at radiofrequency for ranging applications”

Authors: Hussein M. E. Hussein, Seunghwi Kim, Matteo Rinaldi, Andrea Alù, and Cristian Cassella

We thank the Editors for handling our paper and the Referees for their careful reading and their valuable and overall positive comments. Below, we provide a detailed response to all comments from the Referees. We believe to have fully addressed the concerns given by the Referees in the revised manuscript.

Reply to Referee 1

Comment #1. *The manuscript titled "Passive frequency comb generation at radiofrequency for ranging applications" by Hussein M. E. Hussein, Seunghwi Kim, Matteo Rinaldi, Andrea Alù, and Cristian Cassella introduces a new type of RF passive tag called the quasi-Harmonic tag (qHT). This tag generates a frequency comb symmetrically distributed around half the interrogation signal frequency, and the comb's spacing is a function of the interrogation power. The results are interesting and would be beneficial for engineers designing devices for ranging measurements of UV in adversarial environments.*

However, while the theoretical analysis is sound, and the experimental validation is appropriate, I am not convinced that this manuscript provides a clear advance in scientific understanding that would excite the interest of Nature Communications' readership. Therefore, I suggest that the authors target a specialized research journal, such as Nature Microsystems and Nanoengineering or Nature Communications Engineering, to find a more appropriate audience.

To enhance the manuscript's impact, I recommend that the authors answer the following questions: What is new in the presented theoretical analysis that distinguishes it from countless publications studying the nonlinear coupling between a parametric oscillator and high-Q mechanical resonators?

Reply #1. We thank Referee 1 for finding our work “interesting” and “beneficial.” The Referee notices that “the theoretical analysis is sound, and the experimental validation is appropriate.” However, we respectfully disagree with Referee 1 that our work does not provide a clear advance in scientific understanding that would excite the interest of Nature Communications’ readership. We find these comments frustrating, as we cannot find objective evidence to support what is claimed by Referee 1. Nonetheless, we want to further emphasize the impact and novelty of our work in the following:

There exist other few papers studying the coupling between a parametric oscillator and a mechanical resonator, although none discusses the generation of frequency combs through the mechanism exploited for the first time in this work. More generally, as discussed in the main text, there are only few studies treating the generation of frequency combs in the RF domain and these studies are based on: 1) three-wave mixing (Refs. [36-39, 43-44]), 2) nonlinear friction forces (Refs. [40, 41]), and 3) optomechanical/electromechanical interactions (Refs. [35, 42]). While these previous studies successfully showcase RF frequency comb generation, they are primarily designed to provide a demonstration of how to achieve frequency-combs outside the fully-optical framework where frequency combs are generally excited and leveraged. Nevertheless, no prior studies have concretely applied the generation of frequency combs at RF in *practical applications with the purpose of overcoming fundamental limits impacting the performance* of available RF systems. In this regard, our work shows for the first time that frequency combs can be passively generated

by an exceptionally low power RF signal in a passive wireless sensing system, and that such comb-generation allows to remotely and accurately measure a targeted sensing parameter in ways that are intrinsically immune from multipath. Hence, the frequency-comb generation attained and exploited in this work offers a viable solution to circumvent the trade-off between accuracy and tag size in passive wireless sensing systems operating in multipath-rich environments, such as indoor or underground settings. This trade-off results in a significant and inherent reduction of the achievable accuracy for conventional passive tags as the employed tags' size is diminished. While this paper specifically describes how to rely on frequency-combs for ranging, the same technique can be leveraged to remotely monitor any parameter-of-interest in multipath-rich environments. In other words, the device physics we are presenting for the first time in this paper can be leveraged to implement wireless sensing in indoor or underground settings. We can even envision the same concept, once employed for the design of ultrasound transducers and sensors, to allow the extraction of accurate sensing information from highly miniaturized passive sensing nodes deployed inside the human body. The human body, in fact, is one of the most multipath-intense environments wherein acoustic waves can propagate.

With regard to the specific application tackled in this work, this paper provides a clear demonstration of accurate ranging of unmanned vehicles (UVs) through a UHF passive tag able to leverage the reported frequency-comb generation. Our "quasi-harmonic" tags operate passively, enabling far-field ranging measurements with accuracy that largely surpasses any other previous demonstration based on passive tags. Our measurement data, extracted in an uncontrolled electromagnetic setting, confirm that our quasi-harmonic tags enable better accuracies in multipath-rich settings than the current state-of-the-art passive tags operating in an anechoic chamber. Furthermore, our superior ranging accuracy is achieved without requiring multiple tags or multiple interrogating systems, which are often required to achieve ranging information with sub-1 meter-accuracies when employing the existing passive tag technologies for ranging in uncontrolled electromagnetic environments. It is really important to emphasize that attaining a maximum ranging error not exceeding 5 cm for a UV positioned 13 meters away from the interrogating node is absolutely unprecedented. In this regard, Frequency Modulated Continuous Wave (FMCW) radars, notoriously considered the state-of-the-art RF technology for ranging measurements, need interrogation signals with a bandwidth of 3 GHz to be able to achieve similar ranging accuracies than those attained by our quasi-harmonic tag [R1], thereby requiring exceptionally power-hungry and complex transceiver designs. Even more, achieving FMCW radars with such a wide interrogation bandwidth typically demands leveraging carrier frequencies approaching or even exceeding 100 GHz. Obviously, at such mm-wave frequencies much higher EIRP-levels must be transmitted to detect the position of an object from the long distance we have been able to demonstrate in this work, and such high EIRP levels may not even be exploitable according to FCC regulations and due to the existing challenges in designing power-efficient power amplifiers operating within this frequency range. Moreover, it is also worth emphasizing that even LiDARs typically achieve worst ranging accuracies (in the tens of centimeters range) than what we have demonstrated for quasi-harmonic tags in this work [R2]. We strongly believe that all these aspects should not be overlooked while judging the impact of this paper. Many communities are actively engaged in the search for new techniques to accurately measure the position of UVs, driven by the broad and now consolidated awareness that UVs will be key players in the future of the IoT. Another exciting scenario that our technology allows, given the fact that each quasi-harmonic tag only needs a single tone for its interrogation, is the possibility of using signals employed for communication to perform sensing operations, a technique known as "integrated sensing and communication" that has been attracting a growing attention over the last couple of years (Ref. [64]).

Furthermore, it is crucial to highlight that the operation of our system and consequently our analytical model are clearly distinguished from any other works. The generation of combs in our systems does not rely on

any previously discussed mechanism. Instead, it arises from an active pre-synchronization phase in our system, which occurs for driving powers lower than the minimum one needed for the a_1 and b modes to get synchronized and locked to the pump signal. Keep in mind that this is an absolute novelty supported by our theoretical model, as explained in Reply #5 to Referee 2. In other words, differently from prior work, our frequency combs emerge as *pre-synchronized states*. As such, they are triggered by the quasi-periodic motion of a_1 and b , which eventually turns into a synchronized motion for input power levels exceeding the threshold for the activation of the period-doubling regime. This mechanism for comb-generation has never been explored to our knowledge. Also, it comes with a key characteristic that makes it usable for wireless sensing with an exceptional effectiveness: the comb spacing lower monotonically with the pump power. This is a key feature of our technology, as it allows to uniquely correlate any measured line spacing with one specific strength value for the parameter of interest that is monitored.

We have modified the text to highlight better the novelty of our work. We have no doubts that the novelty and experimental results presented in our paper meet the high standards of Nature Communications. We are also sure that our seminal work will attract broad attention within several scientific communities in physics and engineering, including the nonlinear dynamics, the MEMS, the IoT and the wireless communication communities. In our opinion, the dynamics employed in this work for the generation of frequency combs at RF provide a new pathway for physicists and engineers to reduce the energy required for frequency-comb generation in any physical domains.

Comment #2. *How can the equations of motion shown in (1) be rewritten as equation (2), where the mechanical mode b is no longer present? Is the mechanical mode linearly or nonlinearly coupled to the signal mode a_1 ? There seem to be conflicting statements in line 147 and line 203 that require clarification.*

Reply #2. The information regarding the mechanical mode b is included in the effective external stimuli $F(t)$ within Eq. (2), resulting from simply solving Eq. (1). It is worth noting that we have not assumed anything when solving Eq. (1) leading to Eq. (2). The linear response function of the mechanical mode $f(t)$ contains the resonant frequency and loss rate of the b mode, which has been clearly explained in the main text.

We would also like to emphasize that there are no conflicting statements in line 147 and 203, as the Referee claimed. In Eq (1), the a_1 and b modes are ‘linearly’ coupled to each other as described in line 147-148. Meanwhile, in line 203, the microacoustic resonator mainly supporting the b mode is coupled to the a_1 mode.

References:

- [R1] E. Guerrero-Menéndez, "Frequency-modulated continuous-wave radar in automotive applications," 2018.
- [R2] M. Kucharczyk, C. H. Hugenholtz, and X. Y. Zou, "UAV-LiDAR accuracy in vegetated terrain," (in English), J Unmanned Veh Syst, vol. 6, no. 4, pp. 212-234, Dec 2018, doi: 10.1139/juvs-2017-0030.

Reply to Referee 2

Comment #1. *The authors describe a circuit that exploits frequency comb generation for ranging applications. Frequency combs have been extensively studied in optics and the theory and experimental demonstration of the technique described here was performed in references [43,44] using a Josephson junction circuit. The main innovation of the present work consists therefore in demonstrating an application to ranging by exploiting the relationship between the pump power in the circuit and the frequency spacing of the generated harmonics. The technique in particular uses a system of two coupled resonances, one nonlinear and one linear, which is a minimal system to generate frequency combs as detailed in ref [43]. The proposed solution is advantageous compared to linear antennas because it maps the target distance to a frequency difference between two tones, rather than a time delay. This is therefore potentially less susceptible to multipath interference and jamming. I have a few comments:*

Reply #1. We thank Referee 2 for their positive comments, particularly finding our work “advantageous” compared to other linear tag systems and potentially less susceptible to multipath interference and jamming. We would like to answer the rest of comments in the following Replies.

Comment #2. *The weak coupling regime analysis assumes the condition $G < \Gamma/2$. However, the Markovian approximation is technically for $G \ll \Gamma/2$ (much less, not simply less), i.e. for G close to $\Gamma/2$ I would not expect the Markovian approximation to still hold, even if this is still in the weak coupling regime. I think this should be clarified.*

Reply #2. We appreciate the valuable comments by Referee 2, and they are absolutely right. The Markovian approximation for $a(t-\tau) \approx a(t)$ is valid when G is much smaller than Γ . Even in our numerical simulation included in the Supplementary material, we assumed $G = 0.1\Gamma$ which follows the Markovian approximation. Still, the main results do not change regardless of it, since our equation has been fully solved without the Markovian approximation. We have now updated the text for better clarity.

Comment #3. *From the description on page 10 of the circuit in Figure 3a the authors say that $L1, C1$ resonates around $\omega_p/2$, but in the supplementary material, section III.a, they say that it resonates at ω_p . The latter makes more sense to me, since otherwise the pump would not be able to drive the circuit resonantly and the SAW resonator would not be strongly coupled to the electrical resonator at $\omega_p/2$.*

Reply #3. We express our gratitude to the referee for highlighting potential confusion in the preceding sentences. The parametric instability we leverage to generate our frequency-comb hinges on four crucial resonance conditions. To enhance clarity regarding these conditions, we have incorporated an elucidating illustrative figure into Supplementary Figure S5, providing a visual representation of the four prerequisites essential for the parametric circuit's operation.

The figure breaks down the circuit into four distinct branches, labeled 1 through 4. Each branch must exhibit resonance at specific frequencies, as depicted in the corresponding panels of the figure. Notably, we have defined these resonance conditions as follows:

- 1) The combination of $L1 \parallel C1 + L3 + C3$ must series resonate at ω_p
- 2) $L2 \parallel C2$ should parallel resonate at ω_p
- 3) The combination of $L2 \parallel C2 + L3 + C3$ must series resonate at $\omega_p/2$

4) $L_1||C_1$ should parallel resonate at $\omega_p/2$

The following figure graphically summarizes the targeted magnitude of the impedances Z_{1-4} (see figure above) vs. frequency.

Comment #4. *What is the purpose of the third resonator L_3, C_3 ?*

Reply #4. As highlighted in the preceding comment, the L_3-C_3 combination is essential to meet all the resonance conditions that must be satisfied to minimize the activation power for the parametric oscillation in the circuit. A more detailed discussion on these resonance conditions can be found in the supplementary material Section III.a, while their derivation can be found in Ref. [50] of our main paper.

Comment #5. *What is the intuitive reason why I should expect the distance between the comb frequencies to decrease at higher pump power?*

Reply #5. We thank Referee 2 for the valuable comments. Although it is challenging to elaborate on the phenomena ‘intuitively,’ we attempt to answer the question while minimizing the use of mathematical jargon. We believe that our following interpretations can provide intuitive and qualitative explanations, enhancing clarity and understanding.

We can approximate our equations of motion by using a Kuramoto-like model that can exhibit synchronization. Specifically, we can rewrite Eq. (1) in our main text with $a_1 = \alpha e^{i\phi_1}$ and $b = \beta e^{i\phi_2}$. Considering the relative phase between the two modes, $\delta\phi \equiv \phi_1 - \phi_2$, its dynamics may be formulated as:

$$\frac{d\delta\phi}{dt} = 2Gi \sin \delta\phi + i \frac{\Gamma}{2}. \quad (\text{R1})$$

Synchronization occurs for $G > \Gamma/2$, which matches to the condition for comb-generating we have found for our system through the mathematical procedure presented in our paper. In our numerical study in Fig. S4A, the modes a_1 and b are eventually synchronized for high pump powers. The fact that the comb spacing decreases with the pump power is a consequence of the evolution of the motion relative to a_1 and b from a quasi-periodic regime towards their synchronization (Refs. [54,55]). Thus, we confidently assert that our combs are produced during a state of pre-synchronization between a_1 and b . We have added this explanation in the main manuscript and supplementary material.

Comment #6. *The condition for stability limits the maximum value for r_2 . How does this translate in the maximum allowed power? In principle a jammer could be designed that potentially disables the device by sending more power than the stability threshold, thereby reducing the effectiveness of the proposed technique.*

Reply #6. We thank Referee 2 for the valuable comments. The gain saturation parameter r_2 is intricately linked to the pump power $r_2 \approx 2g\sqrt{\kappa_{ex}}a_{in}/\kappa_2$, as elaborated in the supplementary material. Since the pump power is directly proportional to the square of $P_{pump} \propto |a_{in}|^2$, the maximum value of r_2 and the maximum allowed power are interrelated, $P_{th} \propto r_{2,crit}^2$ as indicated in line 130 of the supplement.

Ideally, combs can be generated up to the maximum value of $r_{2,crit}$, as illustrated in Fig. S4A. However, experimental designs may deviate from this ideal scenario due to higher-order nonlinearities in varactors’ capacitance vs. voltage characteristic. $C_{eff}(V) = C_2V^2 + C_3V^3 + \dots$ Nevertheless, we maintain confidence in the efficacy of our technique which shows superior performance over extremely long ranges when compared to other relevant works outlined in Table S3.

With regards to the resilience of our quasi-harmonic tags to jammers, the reviewer is correct. Like for any other passive tags, jamming represents a challenge. It is worth emphasizing that harmonic tags, LiDARs and FMCW radars, all of them used for ranging in the past, are also prone to losses of accuracy in the presence of strong electromagnetic or optical interferers. In fact, like quasi-harmonic tags, they rely on a passive scattering of EM or optical waves on the object whose distance has to be measured. Like any RF system for wireless communication and sensing, immunity to jamming requires additional signal processing that often implies having to use battery-power integrated circuits.

On a separate note, our quasi-harmonic tag requires just a continuous wave input signal to operate. Obviously, this enables a more strategic selection of the operating frequency based on the specific wireless spectral scenario quasi-harmonic tags are designed for. More conclusive and quantitative evaluations of the performance of quasi-harmonic tags in the presence of interferers will be performed in future work.

Comment #7. *In deriving equation S.2 in the supplementary from equation S.1 the authors perform a change of variables to move from the lab frame (where a_1 and a_2 oscillate at frequency ω_1 and ω_2) to a rotating frame (where the variable a oscillates around Δ_a). This point should be clarified in the text in order to avoid confusion.*

Reply #7. We would like to thank Referee 2 for the comments. We have revamped it in the main text for better clarification.

Comment #8. *The authors state after equation S.4 that Δ_b is now assumed to be zero. However the formulas after equation S.5 assume $\Delta_b \neq 0$. I would suggest to clarify at which point in the derivation we should assume $\Delta_b=0$, maybe by moving this sentence to later in the text.*

Reply #8. We understand the potential confusion because of the assumption setting $\Delta_b = 0$ after Eq. (S.4), and now we have moved this sentence to later as suggested.

Comment #9. *Did the authors perform EM liner or nonlinear simulations in the circuit design?*

Reply #9. The circuit design involves running several different simulations such as S-parameters, Harmonic-Balance and Transient. We run the EM simulation to emulate the printed-circuit-board (PCB) impact on the overall circuit impedance. Unfortunately, we could not run nonlinear frequency-domain solutions while using the EM data since HB does not allow the detection of quasi-periodic states. It is also worth pointing out that time-domain simulations also don't work due to the convergence challenges that time-domain algorithms suffer from when dealing with highly nonlinear components, transmission lines and other electromagnetic elements in the circuit.

Comment #10. *What is the measured Q factor of the SAW resonator?*

Reply #10. The measured Q factor of the SAW resonator is 10,900. In fact, the measured Q factor together with other specifications of the utilized SAW resonator are reported in the supplementary material Figure S6.

Comment #11. *The authors indicate that the ranging distance could be extended beyond the 13 meters demonstrated here, which are limited by the threshold power for the harmonic generation to start. In fact this is potentially a limitation compared to passive tags, which do not have a minimum threshold of operation. To understand the applicability of the proposed technique, can the authors comment on how can the distance be increased (and therefore the threshold power decreased)?*

Reply #11. We thank the reviewer for this comment. Linear passive tags have actually a significantly shorter read-range than the quasi-harmonic tag reported in this paper because their backscattered signal is at the

same frequency as their interrogation signal. This poses huge challenges. In fact, the only way for a reader to be able to discriminate the signal coming from a linear passive tag from undesired signals originated from the interrogation signal due to clutter and self-interference is for the tags' components to generate enough delay for the readers to be able to switch its operating mode from transmitting to receiving. Furthermore, even if an exceptionally long-delay is achieved between the interrogation signal and the backscattered signal, the read-range of sensing links that use passive tags is constrained by the limited isolation between readers' TX and RX modules. As a result, the only way to be able to accurately and reliably read the information encoded in the backscattered signal of a linear passive tag is to work close enough to the reader so that the signal-to-noise ratio at the input of the RX module is high enough to ensure a proper reading. On the other hand, nonlinear passive tags like harmonic tags and quasi-harmonic tags completely surpass the performance limitations of linear passive tags, ensuring immunity to self-interference and clutter. Yet, only quasi-harmonic tags allow to accurately extract the sensed information in multipath-rich environments, like indoor or underground settings. When comparing the range of quasi-harmonic tags and harmonic-tags, few considerations can be made:

- 1) Assuming the same interrogation frequency, the path loss that quasi-harmonic tags' output signal undergoes is 12 dB lower than the path loss exhibited by harmonic tags' output signal. This is due to the fact the harmonic tags transmit the sensed information at a 4 times higher frequency than quasi-harmonic tags;
- 2) Attenuation due to propagation of EM signals in indoor or underground environments grows with the operational frequency. Therefore, we expect the harmonic tags' output signal to be more affected by the environment than quasi-harmonic tags.

A detailed comparison between the performance bounds of harmonic tags and subharmonic tags (e.g., tags with backscattered signal having half the frequency of the interrogation signal) is provided in [R1].

With regards to the optimization of the power threshold of quasi-harmonic tags to achieve improved read-ranges with respect to what achieved in this work, we want to bring up a recent paper from our group [R2] where, by applying further design optimizations, we were able to demonstrate an RF power sensitivity for a subharmonic tag of nearly -19 dBm in a 1.3 cm² UHF tag operating at 900 MHz and using only one 80 mm² antenna. Moreover, we would also like to reference another paper about subharmonic tags from our group [R3] where we demonstrated an RF parametric power threshold of 600 nW at 2 MHz. In summary, we strongly believe that further optimizations of the quasi-harmonic tag circuitry and architecture will ensure improved read-ranges in future qHT-prototypes. Additional threshold optimization techniques are discussed in Ref. [50] of our main paper.

References:

- R1] H. M. E. Hussein, M. Rinaldi, M. Onabajo, and C. Cassella, "A chip-less and battery-less subharmonic tag for wireless sensing with parametrically enhanced sensitivity and dynamic range," *Sci Rep-Uk*, vol. 11, no. 1, Feb 12 2021.
- [R2] N. Casilli, L. Colombo, and C. Cassella, "A UHF 1.3 cm² Passive Subharmonic Tag With a 13 m Read-Range," *IEEE Microwave and Wireless Technology Letters*, vol. 33, no. 6, pp. 939-942, 2023.
- [R3] N. Casilli, O. Kaya, T. Kaisar, B. Davaji, P. X. L. Feng, and C. Cassella, "Nonvolatile State Configuration of Nano-Watt Parametric ISING Spins Through Ferroelectric Hafnium Zirconium Oxide MEMS Varactors," presented at the 2023 IEEE 36th International Conference on Micro Electro Mechanical Systems (MEMS), 2023.

Reply to Referee 3

Comment #1. *The paper presents a nonlinear circuit based on lumped elements and MEMS resonators, which is capable of frequency comb generation at microwave frequencies. An important aspect of this circuit is the dependence of the frequency comb spacing on the driving pump power. The authors show an application of the circuit is RFIDs, specifically the determination of the location of a tag from the frequency spacing of the emitted comb. The analysis includes theoretical and experimental results in good agreement with each other. The work is generally interesting and innovative.*

A major issue with the paper is somehow a lack of a cohesive message about its most important and novel result. The paper essentially deals with two topics, the design of circuits for microwave frequency comb generation and the use of such circuits for RFID applications. However, it is not immediately clear to what degree each of these topics advances their respective areas. For example, one of the most exciting aspects of the proposed circuit is the dependence of the comb frequency separation on the pump power. However, it seems that this effect has been theoretically predicted in Ref. 43. An experimental demonstration of this effect is very interesting, but it is not clear whether this is indeed the main contribution of the paper. Furthermore, it is unclear whether this is the first experimental demonstration of this effect or not. The use of MEMS resonators is another interesting aspect of the circuit, but it turns out that their role is auxiliary, since the nonlinearity that is responsible for comb generation is obtained through varactors. In principle, the circuit could be designed with conventional components only by following the architecture in Ref. 43, and the benefit of using MEMS resonators seems to be in controlling the decay rate of one of the cavities. The advantages of the proposed RFIDs from the state of the art are clearer, especially through Table S3 of the SI. However, it is unclear whether they present such a significant advancement to warrant publication in a broad-interest journal instead of a more specialized one. The authors need to spell out better the major contributions of their work and build a case for them.

Other than this general issue, the paper is written very well. The theoretical analysis is extensive and explains well multiple aspects of the proposed circuit. The experimental results are also very interesting and provide solid proof of the concept.

In summary, the paper is interesting, but the authors need to explain better its contribution to determine the suitability of their work for publication in Nature Communications.

Reply #1. We thank Referee 3 for recognizing the "interesting" and "innovative" aspects of our work. The referee has aptly acknowledged the quality of our paper. Furthermore, the comprehensive theoretical analysis effectively elucidates various facets of the proposed circuit. The experimental results, deemed intriguing by the referee, substantiate the conceptual framework with robust evidence. We would like to emphasize our main contribution in this paper again by answering the Referee's questions.

First, we would like to point out the novelty of our model itself compared to other similar works. At glance, it looks like our system has a similar underlying physics with the work in Ref. [43] as the Referee mentioned. In contrast to Ref. [43], there is a notable distinction on the power dependency of the comb spacing. This difference originates from the fundamentally different settings of each model. Let us compare our model with the model in Ref [43] as follows:

$$\left\{ \begin{array}{l} \dot{a}_1 = -\left(\frac{\kappa_1}{2} + i\Delta_a\right)a_1 - iGb - r_1|a_1|^2 a_1 - ir_2 a_1^* \\ \dot{b} = -\left(\frac{\Gamma}{2} + i\Delta_b\right)b - iGa_1 \end{array} \right. \leftrightarrow \left\{ \begin{array}{l} \dot{a} = -\left(\frac{\kappa}{2} + i\Delta_b\right)a - iGb - i\Lambda|a|^2 a, \\ \dot{b} = -\left(\frac{\Gamma}{2} + i\Delta_a\right)b - iGa - i\eta. \end{array} \right. \quad (\text{R2})$$

Our model, represented by the left set of Eq. (R2), differs significantly from the approach presented in Ref. [43] (see the right set of equations). One key distinction lies in the treatment of the driving term: In Ref. [43]'s model, the explicit driving term ($i\eta$) is in the linear mode b , whereas in our model, external pump implicitly drives the system through the nonlinear mode a , acting as both the gain saturation coefficient (r_1) and the small-signal gain (r_2). Notably, the coefficients in our model are inspired from laser physics [Siegman, Lasers, University Science Books (1986)]. From a phenomenological point of view, the works in Refs. [43-44] rely on a three-wave mixing to generate the frequency combs. In contrast, our combs are generated by exploiting the quasi-periodic motion of two pre-synchronized modes. This explains why our comb-spacing *decreases* with the pump power, eventually becoming zero when the a_1 and b modes reach synchronization (see the Reply #5 to Referee 2), while the comb-spacing in Refs. [43-44] has the opposite trend. It is worth pointing out that Ref [43] has not shown the power dependency of the comb spacing while fixing other parameters like the detuning. So, we cited another paper from the same group (Ref [44]) relying on the same model used in Ref. [43]. We firmly believe that our comb-generation mechanism is unique and it differentiates us from any other prior works related to frequency comb generation. In other words, our system is novel and our analytical model and findings have never been investigated before.

Furthermore, previous works on RF frequency comb generation rely on 1) the three-wave mixing, 2) nonlinear friction force, and 3) optomechanical/electro-mechanical interactions. Although these earlier demonstrations effectively highlight the generation of RF frequency combs, their main objective is demonstrating that frequency combs can be generated even outside of the optical domain wherein frequency combs are typically utilized. Differently, we are the first researchers exploring comb generation at RF to overcome the limit of the existing wireless passive sensing systems, providing a path to achieving reliable and accurate data even when exceptionally miniaturized passive sensor tags are used to monitor a targeted parameter of interest.

Regarding the use of the MEMS resonator, this device is not an auxiliary element in the circuit. Instead, it is the key component for the proper operation of our circuit, serving as the fundamental element enabling the existence of frequency-comb states for a wide range of input power levels. As evident from the following figure depicting the response of a circuit simulation we were able to run by using time-domain methods and by analyzing a simplified circuit not including any EM component, the MEMS resonator must be characterized by a Q-value higher than a certain minimum value for the frequency combs to be generated. While this minimum Q value is around 60, a much higher Q is needed to really be able to use qHT for ranging measurements in actual operative scenarios. In fact, the MEMS resonator's Q also dictates the dynamic range of qHTs, which directly translates into the difference between the maximum and the minimum distance that qHTs can cover for a fixed EIRP value used for the interrogation signal. In other words, with a MEMS resonator having a Q of 70, we would have frequency-comb generation at only one power level. Obviously, this prevents from being really able to measure the distance of an item placed within a wide area around the interrogation device. It is also worth emphasizing that replacing the MEMS resonator with a lumped LC resonator would make it impossible to generate the same pre-synchronization states that are responsible for the generation of our frequency combs, regardless of the Q of the LC component. This is due to the relatively low inductance that can be synthesized by using the available

inductor technologies, which has to be limited to few micro Henries (rather than the mHs granted by the MEMS resonator) to ensure that the pump frequency remains higher than the self-resonance frequency of the inductor when operating with a pump signal in the Ultra-High-Frequency range. We have added the following figure together with more detailed discussion in Supplementary Material Section III.E about the importance of the MEMS resonator and the impact of the resonator's Q in governing the operational dynamic range.

As a matter of fact, one of the biggest novelties we are introducing in our work is an innovative technique to circumvent losses of accuracies due to multipath interference when using passive tags for remote sensing. These losses heavily impact the achievable sensing performance when using a passive sensor tag to remotely measure a parameter-of-interest, especially if such tag embodies a small (e.g., omnidirectional) antenna to ensure a large degree of miniaturization. Our quasi-harmonic tags surpass the other available counterparts in both read-range and accuracy. For instance, as described in Reply 1 to Referee 1, the accuracy we are reporting in this paper is better than what achievable when using any other passive tags, and is comparable to or better than what has been demonstrated when using Frequency Modulated Continuous Wave (FMCW) radars and LiDARs [R1, R2]. In this regard, FMCW radars, undoubtedly the most popular RF ranging systems, would require a bandwidth of their chirp interrogation signal approaching 3 GHz (and a carrier frequency in the 30-100 GHz range) to achieve a comparable accuracy to what we have achieved in this work with our quasi-harmonic tag (which requires a single-tone interrogation signal in the Ultra-High-Frequency range). Obviously, having to use a 3 GHz-bandwidth chirp signal centered around 30-100 GHz comes with a huge complexity of the FMCW transceiver, which in turn translates into a huge power consumption that is not compatible with the low-power requirement of most of the existing IoT systems/protocols. Not to mention, having to transmit the interrogation chirp signal around 30-100 GHz translates into required EIRP levels for the interrogating transceiver that may not be allowed according to

FCC regulations and that may be challenging to efficiently achieve considering the low power efficiency of the existing mm-wave power amplifiers.

Moreover, even though this work is specifically focused on the adoption of RF frequency combs for ranging applications, the benefits of our wireless sensing approach are not limited to ranging applications. The ability of our quasi-harmonic tags to provide sensing information immune from multipath renders these systems effective to provide multipath-immune sensing information for any wireless sensing applications. We can even envision the same concept, once employed for the design of ultrasound transducers and sensors, to allow the extraction of accurate sensing information from highly miniaturized passive sensing nodes deployed inside the human body. The human body, in fact, is one of the most multipath-intense environments wherein acoustic waves can propagate. We also want to point out that that future quasi-harmonic tags may incorporate high-Q MEMS resonators not only to activate the desired comb-dynamics but also to achieve high sensing sensitivities to specific parameters of interest, which MEMS devices can do exceptionally well.

We have no doubts that the novelty and experimental results presented in our paper meet the high standards of Nature Communications. We are also sure that our seminal work will attract broad attention within several scientific communities in physics and engineering, including the nonlinear dynamics, the MEMS, the IoT and the wireless communication communities. In our opinion, the dynamics employed in this work for the generation of frequency combs at RF provide a new pathway for physicists and engineers to reduce the energy required for frequency-comb generation in any physical domains.

References:

- [R1] E. Guerrero-Menéndez, "Frequency-modulated continuous-wave radar in automotive applications," 2018.
- [R2] M. Kucharczyk, C. H. Hugenholtz, and X. Y. Zou, "UAV-LiDAR accuracy in vegetated terrain," (in English), *J Unmanned Veh Syst*, vol. 6, no. 4, pp. 212-234, Dec 2018, doi: 10.1139/juvs-2017-0030.

Reviewer #1 (Remarks to the Author):

I have carefully examined the referee reports, the revised manuscript, and the authors' responses to the original feedback. As I mentioned earlier, the paper appears to be technically sound and well-written. The experiments and theoretical modeling also seem to be correct, as far as I can tell.

However, the question remains as to whether this paper should be published in Nature Communications. While I do believe that the results presented in the paper are novel and worthy of publication, I do not think they are groundbreaking enough to be published in such a prestigious journal.

It came to my notice that all the reviewers have noted similarities between the manuscript's results and those reported in other publications, especially in reference 43, which had predicted the same outcomes theoretically. The added paragraphs do a good job of describing the positive engineering aspects of the presented approach, but they do not add anything new regarding the novelty of the existing research.

Reviewer #2 (Remarks to the Author):

I thank the authors for responding to the referees' comments. I think they have adequately addressed my concerns and better clarified the novelty of their work (demonstrating record relative ranging accuracy). I think therefore that the manuscript is suitable for publication.

Reviewer #3 (Remarks to the Author):

The Authors have provided compelling arguments about the significance of their work. Although the differences in the adopted frequency-comb-generation mechanism from other works are probably not significant enough for publication in Nature Communications, the demonstrated range measurement metrics appear to be well beyond the state of the art and could lead to the development of new types of systems for ranging applications. For this reason, I think the paper meets the novelty and impact requirements for publication in Nature Communications.

Response to the Comments from the Referees on “Passive frequency comb generation at radiofrequency for ranging applications”

Authors: Hussein M. E. Hussein, Seunghwi Kim, Matteo Rinaldi, Andrea Alù, and Cristian Cassella

Reply to Referee 1

Comment. *I have carefully examined the referee reports, the revised manuscript, and the authors' responses to the original feedback. As I mentioned earlier, the paper appears to be technically sound and well-written. The experiments and theoretical modeling also seem to be correct, as far as I can tell.*

However, the question remains as to whether this paper should be published in Nature Communications. While I do believe that the results presented in the paper are novel and worthy of publication, I do not think they are groundbreaking enough to be published in such a prestigious journal.

It came to my notice that all the reviewers have noted similarities between the manuscript's results and those reported in other publications, especially in reference 43, which had predicted the same outcomes theoretically. The added paragraphs do a good job of describing the positive engineering aspects of the presented approach, but they do not add anything new regarding the novelty of the existing research.

Reply. We thank Referee 1 for acknowledging the technical soundness of our manuscript, experiments, and theoretical modeling.

In our previous response, we carefully emphasized the impact and novelty of our work particularly highlighting the unique mechanism for the frequency comb generation that is exploited in this work for the first time. Specifically, our model is fundamentally different from others, including Ref 43, as the comb generation in our work stems from pre-synchronization, which is the byproduct of quasiperiodic oscillations. This is in contrast to Ref. 43, where comb is generated from a three-wave mixing. We thoroughly discussed this issue in Reply #1 to Referee 1 and Reply #1 to Referee 3 of the previous response and put this discussion in Supplementary Section II.C, in which we believe the added paragraphs contained new aspects to the paper. The other reviewers agree with our comment, and hopefully Referee 1 will also be convinced. Furthermore, we present compelling experimental evidence demonstrating performance beyond the state-of-the-art in multiple aspects per Referee 3.

In response to your observation regarding the other reviewers, in fact, they all agree that we have adequately addressed their concerns in our previous response, thereby recommending for publication. We hope that the Referee may now appreciate that our paper meets the high standards of Nature Communications in terms of novelty and importance.

Reply to Referee 2

Comment. *I thank the authors for responding to the referees' comments. I think they have adequately addressed my concerns and better clarified the novelty of their work (demonstrating record relative ranging accuracy). I think therefore that the manuscript is suitable for publication.*

Reply. We would like to thank Referee 2 very much for taking the time to review our manuscript and for the positive assessment and recommendation for publication.

Reply to Referee 3

Comment. *The Authors have provided compelling arguments about the significance of their work. Although the differences in the adopted frequency-comb-generation mechanism from other works are probably not significant enough for publication in Nature Communications, the demonstrated range measurement metrics appear to be well beyond the state of the art and could lead to the development of new types of systems for ranging applications. For this reason, I think the paper meets the novelty and impact requirements for publication in Nature Communications.*

Reply. We thank Referee 3 for taking the time to review our paper and for providing valuable feedback. We are grateful for the Referee's recognition of the novelty and potential impact of our work and for the recommendation to publish in Nature Communications.